# Uni4D-LLM: A Unified SpatioTemporal-Aware VLM for 4D Understanding and Generation

## Abstract

Vision-language models (VLMs) have demonstrated strong performance in 2D scene understanding and generation, but extending this unification to the physical world remains an open challenge. Existing 3D and 4D approaches typically embed scene geometry into autoregressive model for semantic understanding and diffusion model for content generation. This paradigm gap prevents a single model from jointly handling both tasks, especially in dynamic 4D settings where spatiotemporal modeling is critical. We propose **Uni4D-LLM**, the first unified VLM framework with spatiotemporal awareness for 4D scene understanding and generation. Our design is guided by two key insights: 1) Unification requires *a shared representation*. We extract semantic and noisy-injected appearance features, incorporate 4D geometric cues, and fuse them into a spatiotemporal-aware visual representation through adaptive cross-attention. 2) Unification requires *a shared architecture*. Both autoregression and diffusion are built on Transformer backbones, and this enables integration into a single LLM with task-specific heads. By aligning visual and linguistic representations, our Uni4D-LLM produces predictions for both understanding and generation within one Transformer-based framework. We further apply instruction fine-tuning on diverse 4D vision-language datasets to improve generalization across tasks. Extensive experiments on multiple benchmarks demonstrate that Uni4D-LLM achieves competitive or superior results compared to state-of-the-art models and offers the first true unification of 4D scene understanding and generation. Our code will be released upon acceptance.

## 1 Introduction

Vision-language models (VLMs) (Alayrac et al., 2022; Liu et al., 2023) have achieved significant progress in scene understanding and generation, but these advances are mainly realized in separate models. In 2D vision, recent works (Fan et al., 2025; Chen et al., 2025) attempt unification by formulating both tasks as next-token prediction. The image patches are discretized into text-like tokens for a large language model (LLM) (Touvron et al., 2023), and task unification is achieved through autoregressive (Wu et al., 2024b) or discrete diffusion (Xie et al., 2024) strategies (*cf.* Fig. 1(a)). Despite being effective for 2D images, these approaches lack explicit spatial and geometric representations and thus cannot generalize to the physical world. To address spatial reasoning, 3D methods (Zhu et al., 2024a; Chen et al., 2024; Zhao et al., 2024; Gao et al., 2024) embed 3D geometry into visual representations. They then use autoregressive models for understanding and diffusion models for generation. Although strong on individual 3D tasks, these methods still treat understanding and generation as separate paradigms.

Extending to spatiotemporal reasoning, 4D approaches (Zhou & Lee, 2025; Zhang et al., 2024a) incorporate temporal cues into 3D geometry. However, they also adopt disjoint solutions: autoregression for understanding and diffusion for generation. Attempts to bridge the gap by coupling LLMs and diffusion models through cross-modal token mapping (Liu et al., 2024a) or geometric-semantic projection (Xu et al., 2025) remain fragmented with separated representation spaces and independent modules. Consequently, no existing framework provides a true unification of 4D scene understanding and generation. This motivates us to propose **Uni4D-LLM**, the first unified VLM framework with spatiotemporal geometry awareness for 4D scene understanding and generation (*cf.* Fig. 1(a–c)).

The design of our Uni4D-LLM is guided by two key insights: 1) Effective unification requires *a shared visual representation*. A 4D scene combines 3D spatial structure with temporal dynamics,

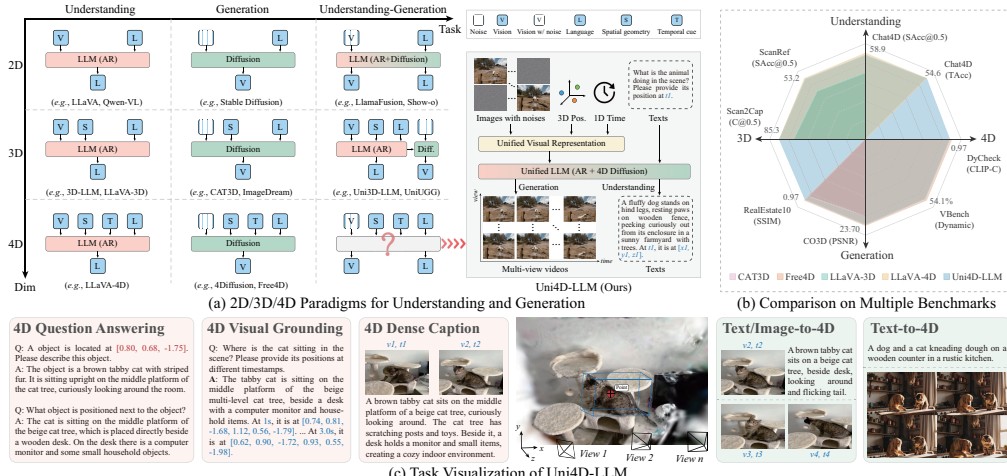

Figure 1: Illustration of 2D/3D/4D paradigms for scene understanding and generation. (a) 2D VLMs unify tasks but lack spatial grounding; 3D/4D extensions add geometry and temporal cues but remain fragmented. Our Uni4D-LLM provides a unified framework for 4D understanding and generation. (b) Benchmark comparison of 3D and 4D paradigms. (c) Results of Uni4D-LLM on diverse tasks.

which demands explicit spatiotemporal modeling. Additionally, scene understanding depends on high-level semantics, and generation reconstructs content from low-level features. These factors motivate a unified visual representation that integrates multi-level features with spatiotemporal awareness. 2) Unification also depends on the ***model architecture***. Although understanding and generation follow different paradigms of autoregression versus denoising diffusion, both are built on Transformer backbones. This structural commonality enables us to unify AR- and diffusion-based reasoning within a single LLM architecture.

As illustrated in Fig. 2, our Uni4D-LLM is built on two key components: a spatiotemporal-aware visual representation and a hybrid LLM architecture. For ***visual representation***, we extract semantic features for understanding and appearance features with injected noise for generation. We further incorporate 1D temporal information into the 3D geometric latent to obtain 4D geometric features. These complementary features are fused through an adaptive cross-attention mechanism to produce a unified visual representation with spatiotemporal awareness. For ***hybrid LLM***, we design a shared Transformer-based architecture that supports both autoregressive reasoning and 4D diffusion denoising. Task distinction is realized through attention masks and multi-task heads, and multi-task predictions are achieved through the alignment of visual and linguistic representations. These designs unify the paradigms of understanding and generation for 4D scenes. Finally, we integrate diverse 4D vision-language datasets and apply instruction fine-tuning to enhance performance on both tasks.

Our main contributions are summarized as follows:

- We present **Uni4D-LLM**, the first general and unified 4D vision-language large multimodal model. Our Uni4D-LLM integrates both scene understanding and generation by combining a spatiotemporal-aware visual representation with a dual-paradigm reasoning architecture.

- We design a unified visual representation with explicit spatiotemporal awareness. An adaptive cross-attention mechanism fuses semantic features, noisy appearance features, and 4D geometric features for stronger multi-task perception on 4D scenes.

- We propose a unified hybrid LLM architecture that supports both autoregressive reasoning and 4D diffusion denoising through attention masks and multi-task heads. This design creates a tighter connection between understanding and generation tasks.

- We incorporate diverse 4D vision-language datasets and apply instruction fine-tuning. Extensive experiments demonstrate that our framework achieves state-of-the-art performance in both 4D scene understanding and generation.

## 2 RELATED WORK

**2D Scene Understanding and Generation.** With the strong reasoning capabilities of large language models (Brown et al., 2020; Touvron et al., 2023), many vision-language models (Liu et al., 2024b;

Lin et al., 2024; Li et al., 2025) have been developed for cross-modal tasks such as understanding and generation. Recently, unifying these two tasks has become an important but challenging direction. Several works (Chen et al., 2025; Wu et al., 2024b; Fan et al., 2025; Xie et al., 2024) address this by formulating both tasks as next-token prediction. They discretize 2D image patches into text-like tokens and feed them into an LLM with a single autoregressive model or a hybrid autoregressive–diffusion model. Although effective for 2D images, these approaches fall short in real-world applications due to the lack of explicit spatial and geometric representations of 3D scenes. This limitation motivates us to enhance visual representations for the physical world, and unifies scene understanding and generation within a single VLM framework.

**3D Scene Understanding and Generation.** A central challenge in extending scene understanding and generation to the physical world is representing spatial characteristics. Existing methods (Deng et al., 2025; Zhu et al., 2024a; Chen et al., 2024; Zhao et al., 2024; Gao et al., 2024) address this by embedding 3D spatial geometry in visual representations. They then adopt autoregressive models for understanding and diffusion models for generation. Although effective for individual tasks, these approaches struggle to unify the two paradigms due to their heterogeneous nature. Some works attempt to pair LLMs with diffusion models through cross-modal token mapping (Liu et al., 2024a) or geometric-semantic conditional projection (Xu et al., 2025). However, these solutions face two major limitations: 1) They are restricted to static scenes and cannot model dynamic temporal variations. 2) Their pipeline-based designs remain disjoint with separated representation spaces and independent trainable modules that prevent true unification. In contrast, we move beyond static 3D and investigate a unified visual representation and model architecture for 4D scene understanding and generation.

**4D Scene Understanding and Generation.** Unlike 3D scenes, 4D scenes require explicit modeling of spatiotemporal characteristics. Existing methods (Zhou & Lee, 2025; Huang et al., 2025; Zhang et al., 2024a; Liang et al., 2024; Wu et al., 2025) follow the 3D paradigm: they embed spatiotemporal geometric features into visual representations, then adopt autoregressive models for understanding and diffusion models for generation. For example, Zhou et al. (Zhou & Lee, 2025) encode 3D positions and 1D time into a learnable spatiotemporal prompt, which is fused with video features and fed into an LLM for 4D scene understanding. However, a unified model architecture for simultaneous 4D scene understanding and generation remains unexplored. We thus take the first step toward such unification. We propose a spatiotemporal-aware visual representation for 4D scenes and integrate autoregressive reasoning with 4D diffusion into a single LLM architecture to bridge understanding and generation within one framework.

## 3 OUR UNI4D-LLM

**Overview.** Fig. 2 shows the architecture of our Uni4D-LLM. Given a (multi-view) video sequence, our Uni4D-LLM unifies 4D scene understanding and generation through the following three stages:

1) **Unified Spatiotemporal-Aware Visual Representation (*cf.* Sec. 3.1).** We construct unified task representations through 4D scene modeling. A video sequence is encoded into visual and geometric latents. The visual latent $z_v$ produces semantic features $f_s = \text{SE}(z_v)$ for understanding and appearance features: $f_a = \text{NE}(z_v + \epsilon)$ with injected noise $\epsilon$ for generation. The geometric latent $z_{3D}$ is combined with time $t$ to produce 4D geometric features $f_{4D} = \text{STE}(z_{3D}, t)$, where $\text{SE}(\cdot)$, $\text{NE}(\cdot)$, and $\text{STE}(\cdot)$ denote semantic, noise, and spatiotemporal embeddings, respectively. These features are fused through adaptive cross-attention to produce a unified spatiotemporal-aware visual representation: $f_v^{4D} = \text{Ad\_CAtt}([f_s, f_a], f_{4D}, f_{task})$, which is dynamically modulated by the task prompt $f_{task}$ to differentiate between understanding and generation.

2) **Unified Hybrid LLM Architecture (*cf.* Sec. 3.2).** We unify different task paradigms within a single Transformer backbone $\mathcal{T}(\cdot)$. Given input $x^{in}$ and attention mask $\mathcal{M}$, we obtain hidden features $h = \mathcal{T}(x^{in}, \mathcal{M})$. An autoregressive head produces $y^{AR} = \mathcal{H}_U(h)$ for understanding, and a diffusion head outputs $y^{Diff} = \mathcal{H}_G(h)$ for generation. The unified hybrid architecture is thus written as LLM : $\{\mathcal{T}; \mathcal{M}\} \mapsto \{y^{AR}, y^{Diff}\}$, where $\mathcal{M}$ is a predefined attention mask to control the task-specific information flow.

3) **Multimodal Alignment and Multi-Task Optimization (*cf.* Sec. 3.3).** We align visual and linguistic representations to enable joint optimization. The unified visual representation is projected into the language embedding space: $\tau_v^{4D} = \text{Proj}(f_v^{4D})$, where $\tau_v^{4D}$ is aligned with linguistic tokens $\tau_l$. The unified LLM consumes these tokens to produce multi-task outputs, which are decoded into texts and multi-view images/videos. Training is guided by joint objectives for both understanding and generation.

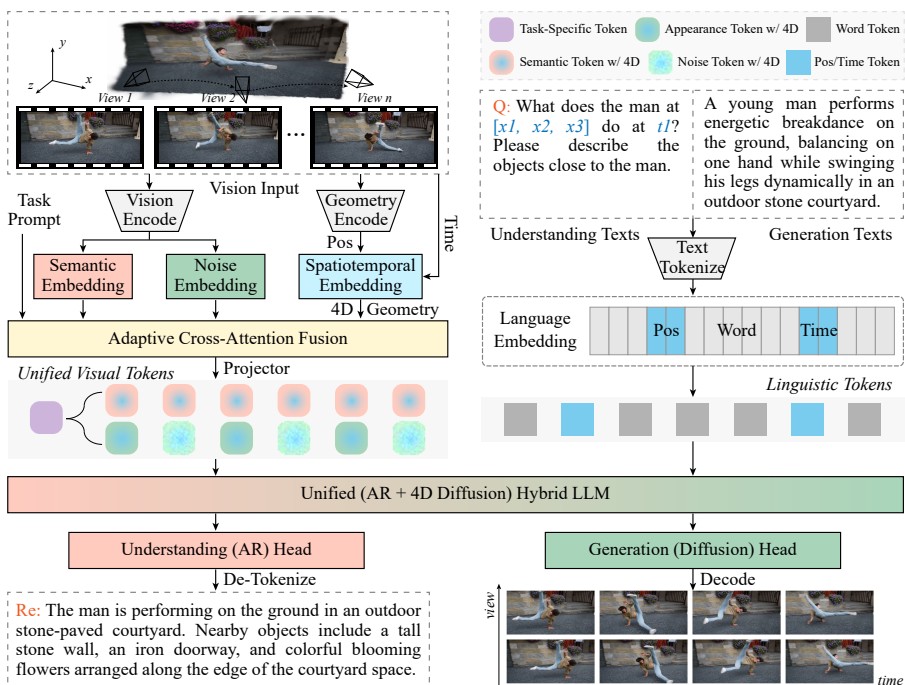

Figure 2: Uni4D-LLM unifies 4D scene understanding and generation via three stages: 1) **Unified visual representation.** Equip unified task representations with 4D scene modeling via adaptive cross-attention fusion. 2) **Unified model architecture.** Integrate various task paradigms into a single LLM. 3) **Multi-task optimization.** Achieve multimodal alignment and joint task optimization.

**Remarks.** The output texts and images/videos denote the results of 4D scene understanding and generation, respectively. Our framework unifies 4D scene understanding and generation from three perspectives: visual representation, model architecture, and task optimization.

## 3.1 UNIFIED SPATIOTEMPORAL-AWARE VISUAL REPRESENTATION

4D VLMs face two key challenges in visual representation for 4D scene understanding and generation: heterogeneous task representation and 4D scene modeling. In this section, we construct multi-task features and model the spatiotemporal characteristics of the scene, and further integrate them into a unified visual representation for both 4D scene understanding and generation.

**Task-Specific Visual Representation.** The task representations for understanding and generation differ fundamentally. Understanding requires modeling contextual and logical knowledge, while generation focuses on reconstructing visual content. This distinction motivates us to adopt a divide-and-conquer strategy for feature modeling. Consider multi-view videos as an example. We first employ a VAE (Kingma & Welling, 2013) as the vision encoder to map video sequences into a visual latent $z_v$ indexed by view and time. For the understanding task, we use SigLIP (Zhai et al., 2023) as the semantic embedding SE and fine-tune it to extract high-level semantic features from the visual latent, i.e. $f_s = \text{SigLIP}(z_v)$. For the generation task, we design a noise embedding NE with a linear layer and a noise scheduler. The linear layer maps the visual latent to the appearance features, and the noise scheduler randomly injects noise with varying intensity. Formally, the appearance features are given by $f_a = (1 - m) \odot W_a z_v + m \odot (W_a z_v + \alpha_t \epsilon)$, where $m$ is a random mask, $\alpha_t$ is the step-dependent noise intensity, and $\epsilon \sim \mathcal{N}(0, I)$ is Gaussian noise.

**Spatiotemporal Geometric Representation.** Unlike 2D scenes, effective reasoning in 4D VLMs requires explicit spatiotemporal awareness. For the spatial dimension, we adopt MonST3R (Zhang et al., 2024b) as a geometry encoder for dynamic scenes. It transforms video sequences into a geometric latent: $z_{3D} = \{z_{3D}^{pose}, z_{3D}^{posi}\}$, which captures both camera poses and 3D scene positions. For the temporal dimension, we extract timestamps from the videos. These are encoded using a Fourier-based strategy $\mathcal{F}(\cdot)$ (Li et al., 2021), which converts time into learnable feature patterns. Finally, we introduce a spatiotemporal embedding STE that concatenates the geometric 3D features

with the encoded 1D time and maps them into a 4D representation through a linear layer: $f_{4D} = W_{4D} \cdot [z_{3D} \parallel \mathcal{F}(t)]$. The resulting $f_{4D}$ represents the spatiotemporal characteristics of the scene.

**Adaptive Cross-Attention Fusion.**
We propose an adaptive cross-attention mechanism to achieve a unified representation that supports multiple tasks and captures 4D scene structure. As illustrated in Fig. 3, this mechanism fuses semantic features, appearance/noise features, and 4D geometric features. The fusion process consists of two steps: task fusion and 4D fusion. In task fusion, we introduce a task

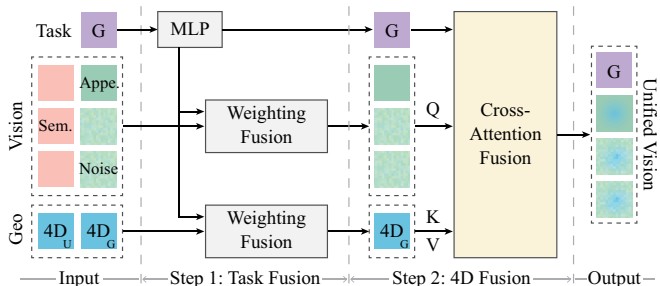

Figure 3: Illustration of adaptive cross-attention fusion.

prompt $f_{task}$ that distinguishes between understanding and generation. It serves as a modulation parameter that dynamically balances semantic and appearance/noise features through weighted fusion: $\hat{f}_v = \alpha f_s + (1 - \alpha) f_a$, where $\alpha = \text{MLP}(f_{task})$. In 4D fusion, task-specific visual features $\hat{f}_v$ are used as queries. We adaptively combine geometric information by weighting spatial positions and poses as: $\hat{f}_{4D} = \alpha f_{4D}^{posi} + (1 - \alpha) f_{4D}^{pose}$. We then apply cross-attention, where $q = w_q \hat{f}_v$, $k = w_k \hat{f}_{4D}$, and $v = w_v \hat{f}_{4D}$. The fusion output is computed as $f_{uni} = \text{softmax}(qk^\top / \sqrt{d}) v$. Finally, the unified visual representation is obtained as $f_v^{4D} = \{\alpha, f_{uni}\}$. This representation integrates the task prompt with task-specific visual features and enriches them with 4D spatiotemporal geometry.

## 3.2 UNIFIED HYBRID LLM ARCHITECTURE

Although the unified visual representation encompasses task-specific features for both understanding and generation in 4D scenes, the modeling paradigms of these two tasks are fundamentally different. This raises an important question: "*Can a unified architecture be devised to simultaneously learn heterogeneous task paradigms?*"

**Shared Transformer Backbone.** Both understanding and generation baselines (Wu et al., 2024b; Zhang et al., 2024a) are built on similar Transformer structures. However, instead of concatenation of separate task-specific models with different weights, the key to unification is a single Transformer that shares weights across tasks. To this end, we construct a shared Transformer as the LLM backbone as shown in Fig. 4(a). Additionally, we design an autoregressive head for understanding and a diffusion head for generation. The shared backbone follows a standard Transformer block that includes multi-head self-attention, a feed-forward network, and layer normalization with residual connections. We introduce predefined attention masks to account for paradigm differences between tasks. These masks dynamically control the information flow within each block for the same backbone to adapt to different tasks. Finally, multiple blocks are stacked to form the shared Transformer backbone $\mathcal{T}$.

**Autoregressive Model.** The autoregressive model is typically a feed-forward LLM designed for classification-based prediction. This motivates us to adopt the shared Transformer backbone $\mathcal{T}$ as the vision-language model, and append an understanding head composed of a linear layer and a softmax layer. The output is given by $y^{AR} = \text{Softmax}(W_u \cdot \mathcal{T}(x_{in}, \mathcal{M}_{\mathcal{U}}))$. For the predefined mask $\mathcal{M}_{\mathcal{U}}$ in Fig. 4(b), we configure three types of attention: 1) Full attention across all visual tokens for global contextual association; 2) Asymmetric attention between linguistic and visual tokens for conditional autoregressive understanding; 3) Causal attention among linguistic tokens.

**4D Diffusion Model.** The generative model is typically formulated as an iterative denoising diffusion process, which can be viewed as a regression-based fitting model. To this end, we reuse the shared Transformer backbone $\mathcal{T}$ to simulate the multi-step diffusion process and append an MLP-based generation head. The process is defined as: $\hat{\epsilon}^{(t)} = \text{MLP}(\mathcal{T}(x^{(t)}, \mathcal{M}_{\mathcal{G}}))$ with initialization $x^T = x_{in}$ and denoising iteration $x^{(t-1)} = x^{(t)} - \alpha_t \hat{\epsilon}^{(t)}$ for $t \in \{1, \ldots, T\}$, where $T$ is the total number of diffusion steps. The final output is $y^{Diff} = x^{(0)}$. For the predefined mask $\mathcal{M}_{\mathcal{G}}$ in Fig. 4(c), we adopt a spatiotemporal alternating attention strategy, *i.e.* view–time–view sequence to ensure spatial consistency and temporal continuity. At the view level with time fixed, we configure: i) Full attention among noisy visual tokens from different views; ii) Full attention between noise-free visual tokens and linguistic tokens for multimodal association; iii) Asymmetric attention between noisy tokens and

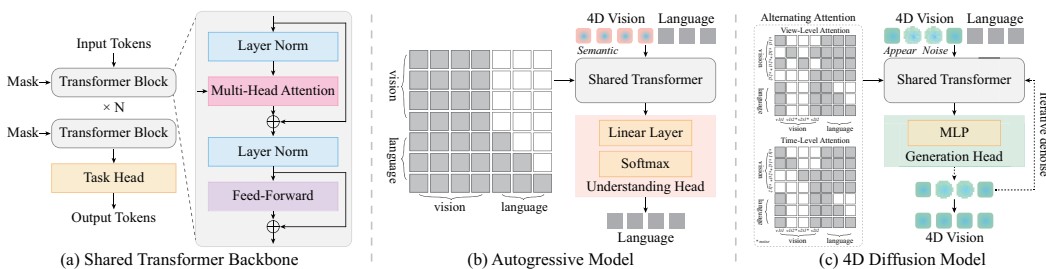

Figure 4: Architecture of unified hybrid LLM. (a) Shared transformer serves as the LLM backbone to support (b) autoregressive model and (c) 4D diffusion model via attention masks and task heads.

noise-free/linguistic tokens for conditional generation; iv) causal attention among linguistic tokens. Time-level attention follows the same design as view-level attention.

### 3.3 MULTIMODAL ALIGNMENT AND MULTI-TASK OPTIMIZATION

The unified visual representation and the unified LLM architecture essentially integrate understanding and generation tasks for 4D scenes at the levels of feature modeling and model inference. However, the simultaneous optimization of these two tasks remains a critical challenge.

**Vision-Language Alignment.** Considering that the input of a large language model requires text-like tokens, we first introduce a multi-layer perceptron as the projection function $\text{Proj}(\cdot)$ to map the unified visual representation into the language embedding space: $\tau_v^{4D} = \text{MLP}(f_v^{4D})$. Next, we tokenize the input instruction into the language embedding space to obtain the linguistic tokens $\tau_l$, where the textual position and time are enhanced by a special token embedding (Li et al., 2025). In this way, we ensure a preliminary alignment between visual and linguistic representations.

**Joint Optimization.** We concatenate the unified visual and linguistic tokens and feed them into the hybrid LLM. Guided by the task prompt, the model uses the autoregressive pathway for linguistic outputs and the 4D diffusion pathway for visual outputs. For optimization, we apply cross-entropy loss on predicted linguistic tokens: $\mathcal{L}_{AR} = -\sum_i \log \hat{p}_\theta(\hat{\tau}_{l,i} \mid \tau_{l,<i}, \tau_v^{4D})$, and MSE loss on predicted noise: $\mathcal{L}_{Diff} = \mathbb{E}_t\left[\|\hat{\epsilon}^{(t)} - \epsilon\|^2\right]$, which is applied only to tokens derived from noisy inputs. The total objective: $\mathcal{L} = \lambda_{AR}\mathcal{L}_{AR} + \lambda_{Diff}\mathcal{L}_{Diff}$. Finally, linguistic tokens are de-tokenized into text, and visual tokens are decoded into multi-view/time images. Optionally, Gaussian Splatting (GS) (Kerbl et al., 2023) can be applied to refine the details of the image.

## 4 TRAINING PIPELINE

To ensure the stability of the training process and improve the performance of our model for 4D understanding and generation, we divide the entire training process into three stages as follows:

**Stage 1: Fundamental Representation Learning.** This stage equips the model with multi-task visual and linguistic representations using large-scale 2D image/video–text datasets for captioning (ImageNet-1K (Deng et al., 2009), WebVid-10M (Bain et al., 2021)) and visual QA (GranD (Rasheed et al., 2024), ANet-RTL (Huang et al., 2024a)). Captions also serve as conditional text for scene generation to align visual and linguistic features as the foundation for both tasks. We update the embeddings, projector, lower LLM layers, and multi-task heads, and freeze the remaining modules.

**Stage 2: Multimodal Spatiotemporal Alignment.** This stage enhances spatiotemporal awareness and adapts the model to the physical world. We use 3D scene understanding datasets for captioning, QA, and grounding (Scan2Cap (Chen et al., 2021), ScanQA (Azuma et al., 2022), ScanRef (Chen et al., 2020)), a small 4D captioning dataset (Chat4D (Zhou & Lee, 2025)), and 3D generation datasets (CO3D (Reizenstein et al., 2021), RealEstate10k (Zhou et al., 2018)). These hybrid datasets align fine-grained spatiotemporal information across modalities. We update the spatiotemporal embedding, adaptive cross-attention fusion, higher LLM layers, multi-task heads, and freezing other modules.

**Stage 3: 4D Task Instruction Fine-Tuning.** This stage improves generalization to complex 4D scene understanding and generation. We use 4D vision-language datasets (Chat4D (Zhou & Lee, 2025), DyCheck (Gao et al., 2022)) and apply instruction fine-tuning to adapt the model to dynamic 4D environments. All trainable parameters are optimized with LoRA adapters (Hu et al., 2022), and the vision encoder-decoder and geometry encoder remain frozen.

Table 1: Quantitative results for scene understanding tasks on different 3D and 4D datasets.

| | Methods | 3D Benchmark | | | | | | | | 4D Benchmark | | | | |
| | | Scan2Cap | | | ScanQA | | | Multi3DRefer | ScanRef | Chat4D | | | | |
| | | C@0.5↑ | B-4@0.5↑ | M@0.5↑ | C↑ | B-4↑ | M↑ | F1@0.5↑ | SAcc@0.5↑ | C↑ | B-4↑ | M↑ | SAcc@0.5↑ | TAcc↑ |
|---|---|---|---|---|---|---|---|---|---|---|---|---|---|---|
| 3D | 3D-LLM | – | – | – | 69.4 | 12.0 | 14.5 | – | – | 61.6 | 11.5 | 12.3 | 31.4 | – |
| | Chat-3D v2 | 63.9 | 31.8 | – | 87.6 | 14.0 | – | 41.6 | 38.4 | 81.8 | 13.7 | – | 39.5 | – |
| | 3D-LLaVA | 78.8 | 36.9 | 27.1 | 92.6 | 17.1 | 18.4 | – | – | 85.1 | 16.0 | 18.2 | 52.0 | – |
| | PQ3D | 80.3 | 36.0 | 29.1 | 87.8 | – | 17.8 | 50.1 | 51.2 | 84.7 | 14.3 | 17.5 | 51.5 | – |
| | LLaVA-3D | 79.2 | 41.1 | 30.2 | 91.7 | 14.5 | 20.7 | – | 42.2 | 87.4 | 14.8 | 19.4 | 45.6 | – |
| | Video-3D LLM | 83.8 | 42.4 | 28.9 | **102.1** | 16.2 | 19.8 | 52.7 | 51.7 | 89.4 | 16.1 | 19.2 | 52.8 | – |
| 4D | LLaVA-4D | **85.3** | **45.7** | **31.3** | 97.8 | **17.9** | 21.2 | **54.3** | **53.2** | 93.5 | **17.2** | **21.0** | **58.9** | **54.6** |
| | Uni4D-LLM (Ours) | 85.1 | 45.4 | 31.0 | 100.5 | 17.4 | **21.2** | 53.9 | 53.0 | **93.8** | 17.1 | 20.6 | 58.2 | **54.6** |

Table 2: Quantitative results for scene generation tasks on different 3D and 4D datasets.

| | Methods | 3D Benchmark | | | | | | 4D Benchmark | | | | | | | |
| | | CO3D | | | RealEstate10 | | | DyCheck | | | | VBench | | | |
| | | PSNR↑ | SSIM↑ | LPIPS↓ | PSNR↑ | SSIM↑ | LPIPS↓ | PSNR↑ | LPIPS↓ | FVD↓ | CLIP-C↑ | Cons↑ | Dyn↑ | Aes↑ | T-Ali↑ |
|---|---|---|---|---|---|---|---|---|---|---|---|---|---|---|---|
| 3D | 3D-GS | 19.28 | 0.61 | 0.54 | 22.65 | 0.76 | 0.35 | 12.70 | 0.55 | – | – | – | – | – | – |
| | ImageDream | 21.95 | 0.71 | 0.35 | 29.87 | 0.94 | 0.10 | 15.26 | 0.43 | – | – | 88.3% | – | 49.2% | 21.5% |
| | D-Craft3D | 20.35 | 0.68 | 0.42 | 27.74 | 0.90 | 0.17 | 15.52 | 0.44 | – | – | 87.4% | – | 48.0% | 20.6% |
| | CAT3D | 22.79 | 0.73 | 0.30 | 31.07 | 0.95 | 0.09 | – | – | – | – | – | – | – | – |
| 4D | 4D-GS | 19.57 | 0.65 | 0.52 | 22.70 | 0.79 | 0.33 | 16.54 | 0.35 | 462.5 | 0.89 | – | – | – | – |
| | 4D-fy | 22.62 | 0.71 | 0.30 | 28.11 | 0.91 | 0.15 | 17.92 | 0.31 | 255.2 | 0.92 | 91.6% | 53.3% | 54.5% | 25.7% |
| | 4Diffusion | 23.55 | 0.79 | 0.24 | 31.62 | 0.95 | 0.08 | 20.36 | 0.19 | 182.7 | 0.96 | 94.5% | 53.6% | 57.2% | 25.8% |
| | Free4D | **23.70** | **0.81** | **0.22** | **31.90** | **0.97** | **0.07** | **21.55** | **0.16** | **140.6** | **0.97** | **96.9%** | **54.1%** | **61.9%** | 26.0% |
| | Uni4D-LLM w/o GS | 23.04 | 0.75 | 0.26 | 29.94 | 0.94 | 0.10 | 20.23 | 0.20 | 197.1 | 0.96 | 94.1% | 53.7% | 57.8% | 25.9% |
| | Uni4D-LLM w/ GS | 23.61 | 0.80 | **0.22** | 31.75 | 0.96 | **0.07** | 21.38 | 0.17 | 152.3 | **0.97** | 96.5% | 53.9% | 61.1% | **26.2%** |

## 5 EXPERIMENTS

**Implements Details.** Our Uni4D-LLM model utilizes the pre-trained weights of Qwen2.5-7B-Instruct (Bai et al., 2025), the vision encoder-decoder of VAE proposed in Wan2.1 (Wan et al., 2025) and the geometry encoder of MonST3R (Zhang et al., 2024b). The adaptive cross-attention fusion module is a Transformer-based network architecture. The whole model is trained on 8 RTX 4090 GPUs using AdamW as the optimizer. In training stage 1, we set the learning rate to $1.0e-4$ with a batch size of 16. In training stage 2, we set the learning rate to $5.0e-5$ with a batch size of 8. We use a learning rate of $1.0e-5$ with a batch size of 12 in training stage 3.

**Comparison Methods.** Since Uni4D-LLM is a multi-task model for 4D scenes, we construct a comprehensive set of baselines across both task types: understanding and generation, and scene dimensions: 3D and 4D. For scene understanding, we compare with 3D VLMs including 3D-LLM (Hong et al., 2023), Chat-3D v2 (Huang et al., 2023), 3D-LLaVA (Deng et al., 2025), PQ3D (Zhu et al., 2024b), LLaVA-3D (Zhu et al., 2024a), and Video-3D LLM (Zheng et al., 2024), as well as the 4D VLM LLaVA-4D (Zhou & Lee, 2025). For scene generation, we compare against 3D diffusion models including ImageDream (Wang & Shi, 2023), DreamCraft3D (Sun et al., 2023), and CAT3D (Gao et al., 2024); 4D diffusion models including 4D-fy (Bahmani et al., 2024), 4Diffusion (Zhang et al., 2024a), and Free4D (Liu et al., 2025); and Gaussian splatting models in both 3D and 4D, including 3D-GS (Kerbl et al., 2023) and 4D-GS (Wu et al., 2024a).

**Evaluation Metric.** We compare methods on multiple 3D&4D understanding and generation benchmarks. For understanding, we evaluate on Scan2Cap (Chen et al., 2021), ScanQA (Azuma et al., 2022), ScanRef (Chen et al., 2020), Multi3DRefer (Zhang et al., 2023), and Chat4D (Zhou & Lee, 2025), using CiDEr (C), BLEU-4 (B-4), METEOR (M), F1 for the quality of text response, and spatial/temporal IoU grounding accuracy (S/TAcc). For generation, we adopt CO3D (Reizenstein et al., 2021), RealEstate10 (Zhou et al., 2018), DyCheck (Gao et al., 2022), and VBench (Huang et al., 2024b), reporting PSNR, SSIM, LPIPS for spatial consistency, FVD for video quality, CLIP-C for temporal consistency, and VBench metrics (Consistency, Dynamic Degree, Aesthetic, Text Alignment). Experiments in Sec. 5.2 are conducted on 4D datasets.

### 5.1 COMPARISON WITH STATE-OF-THE-ART MODELS

**Quantitative Results on Understanding.** Table 1 reports comparisons between 3D and 4D VLMs on both 3D and 4D understanding tasks. Our Uni4D-LLM consistently surpasses most 3D methods and achieves performance on par with several state-of-the-art models. We demonstrate clear temporal advantages over 3D VLMs on 4D benchmarks and remain broadly competitive with the latest 4D

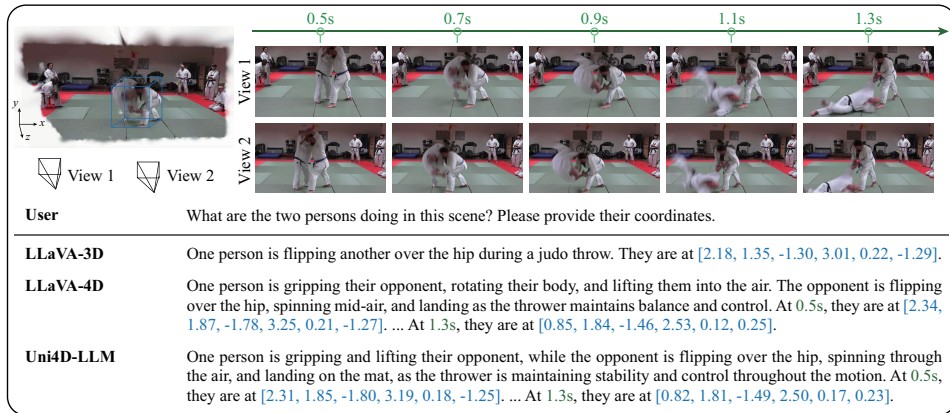

Figure 5: Visual comparison on 4D scene understanding.

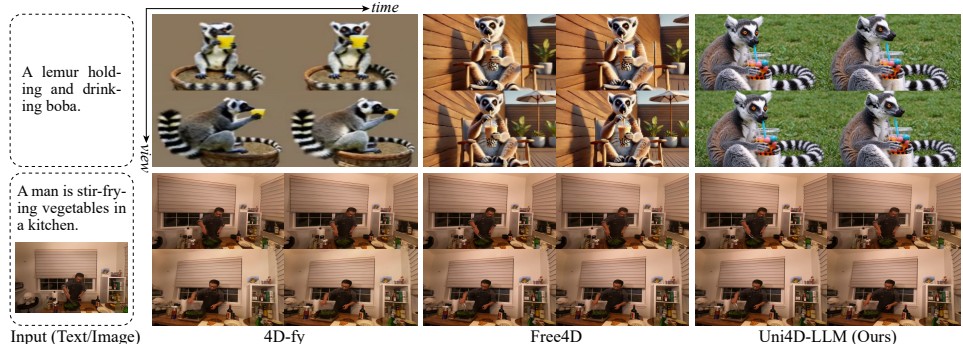

Figure 6: Visual comparison on 4D scene generation, *e.g.*, text-to-4D and text/image-to-4D.

VLMs with only minor gaps on a few metrics. These results confirm the strong effectiveness and overall superiority of Uni4D-LLM across diverse benchmarks.

**Quantitative Results on Generation.** In Table 2, we present comparisons with 3D and 4D diffusion models, and 3D and 4D Gaussian splatting (GS) models. We also evaluate GS as a post-processing strategy for our framework. Our Uni4D-LLM outperforms most existing diffusion and GS models on both 3D and 4D generation tasks. Without GS, it performs slightly below the latest 4D diffusion models equipped with GS. However, our model achieves comparable or superior results on several metrics when combined with GS. Generally, our Uni4D-LLM delivers strong generation performance in both 3D and 4D settings, and GS further enhances the visual detail and quality of the outputs.

**Qualitative Results.** Figures 5 and 6 show representative 4D scenes comparing Uni4D-LLM with both 3D and 4D models. In 4D understanding, 3D VLMs struggle to capture temporal dynamics, while our model demonstrates strong spatiotemporal reasoning on par with recent 4D VLMs. In 4D generation, our Uni4D-LLM produces sharp and coherent results that rival those of advanced 4D diffusion models. These results demonstrate the superiority of Uni4D-LLM in 4D understanding and generation, underscoring its potential as a unified multi-task framework for the physical world.

## 5.2 ABLATION STUDY AND DISCUSSION

**Role of Spatiotemporal Embedding.** Table 3 analyzes the impact of spatiotemporal embedding on 4D understanding and generation. The model maintains reasonable performance on several understanding metrics upon removal of the embedding, but fails at fine-grained reasoning

Table 3: Impact of spatiotemporal embedding.

| Embedding | Chat4D | | | DyCheck | | |
|---|---|---|---|---|---|---|
| type | C↑ | SAcc@0.5↑ | TAcc↑ | PSNR↑ | FVD↓ | CLIP-C↑ |
| w/o Embedding | 75.3 | 12.4 | 10.4 | 19.40 | 260.1 | 0.93 |
| w/ Spatial | 91.0 | 56.5 | 10.4 | 20.95 | 213.6 | 0.94 |
| w/ SpatioTemp. | **93.8** | **58.2** | **54.6** | **21.38** | **152.3** | **0.97** |

and suffers degraded generation quality. Spatial embedding improves spatial understanding and fidelity, and temporal embedding enhances temporal reasoning and generative consistency.

**Effectiveness of Unified Representation & Architecture.** Figure 7 analyzes the impact of unified representation and architecture on 4D understanding and generation. Models with separate represen-

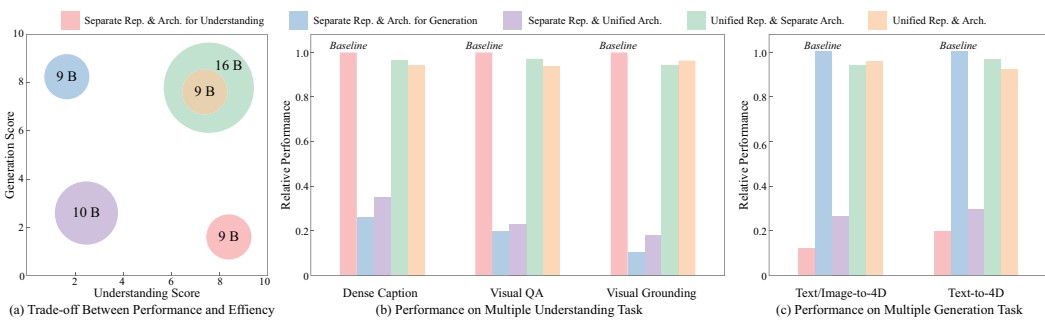

Figure 7: Effectiveness of unified representation and architecture. The understanding and generation scores are obtained by weighted aggregation of the corresponding task-specific normalized metrics.

tations and architectures perform well on only single tasks. Combining separate representations with a unified architecture degrades performance due to feature mismatch. Unified representations with separate architectures improve results but incur large parameter costs. Unified representation and architecture achieve strong multi-task performance without excess parameters.

**Choice of Visual Representation Fusion Strategy.** Table 4 compares fusion strategies for visual representation. Attention-based fusion outperforms concatenation and weighting, which rely on fixed global weights and ignore task-specific differences. In contrast, attention-based fusion adaptively balances task-specific and 4D features for stronger multi-task modeling.

Table 4: Choice of representation fusion strategy.

| Fusion strategy | Chat4D | | | DyCheck | | |
|---|---|---|---|---|---|---|
| | C↑ | SAcc@0.5↑ | TAcc↑ | PSNR↑ | FVD↓ | CLIP-C↑ |
| Concat | 88.4 | 54.1 | 51.4 | 20.75 | 185.4 | 0.95 |
| Weighting | 89.6 | 54.5 | 52.0 | 21.02 | 169.1 | 0.96 |
| Attention | **93.8** | **58.2** | **54.6** | **21.38** | **152.3** | **0.97** |

**Importance of Attention Mask.** We evaluate the role of the attention mask in our unified model. As shown in Table 5, it significantly improves both understanding and generation performance. The attention mask works by dynamically controlling and modulating the information flow based on the task setting, which enables more effective reasoning across different paradigms.

Table 5: Discussion on attention mask.

| Attention for LLM | Chat4D | | | DyCheck | | |
|---|---|---|---|---|---|---|
| | C↑ | SAcc@0.5↑ | TAcc↑ | PSNR↑ | FVD↓ | CLIP-C↑ |
| w/o Mask | 89.4 | 53.9 | 50.8 | 19.81 | 232.4 | 0.93 |
| w/ Mask | **93.8** | **58.2** | **54.6** | **21.38** | **152.3** | **0.97** |

**Impact of Spatiotemporal Alternating Strategy.** Table 6 compares different attention mask sampling strategies for 4D generation. The overall performance is acceptable without sampling, but the spatial and temporal consistency remain weak. View-only sampling improves spatial coherence and time-only sampling strengthens temporal continuity. Our spatiotemporal alternating strategy achieves the best results across all metrics with both spatial fidelity and temporal stability.

Table 6: Role of attention sampling for generation.

| Sampling strategy | PSNR↑ | FVD↓ | CLIP-C↑ |
|---|---|---|---|
| w/o Sampling | 20.27 | 194.2 | 0.95 |
| w/ View-only | 20.95 | 187.9 | 0.95 |
| w/ Time-only | 20.86 | 161.3 | 0.96 |
| w/ Alternating | **21.38** | **152.3** | **0.97** |

**Limitation.** Despite strong performance in short-term scene understanding and generation, our Uni4D-LLM struggles with long-term dynamics. Capturing such variations requires memory-based reasoning to model cross-spatiotemporal interactions and causal relations. For future work, we plan to integrate a world model (Ha & Schmidhuber, 2018) to enable long-term spatiotemporal reasoning and extend scene understanding and generation to longer temporal horizons.

## 6    CONCLUSION

In this work, we introduce Uni4D-LLM, the first general vision–language model that unifies 4D scene understanding and generation. Our framework builds a spatiotemporal-aware visual representation for multi-task 4D perception and designs a hybrid LLM architecture that supports both autoregressive and 4D diffusion models to bridge understanding and generation. Through multimodal alignment between visual and linguistic representations, our unified LLM produces effective multi-task predictions under joint optimization. By integrating visual representation, model architecture, and task optimization, our Uni4D-LLM achieves a comprehensive unification of 4D scene understanding and generation. We further fine-tune on diverse 4D vision-language datasets and validate the effectiveness of our approach through extensive experiments. We believe that this work paves the way toward unified multi-task multimodal models for the physical world.

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
