# Uni4D-LLM: A Unified SpatioTemporal-Aware VLM for 4D Understanding and Generation
## —*Supplementary Material*—

In this supplementary material, we provide additional technical details and experimental validations on our training details, method, and comparison. For our training details, we introduce the detailed training pipeline and datasets in Sec. 1. For our method, we conduct extensive experiments on ablation study and discussion in Sec. 2, including applicability to different camera settings in Sec. 2.1, task prompt in Sec. 2.2, textual coordinate embedding in Sec. 2.3, and the potential solution to the limitations in Sec. 2.4. Regarding our comparison, we provide more quantitative and qualitative results of 2D/3D/4D scene understanding and generation in Sec. 3.1-3.3.

## 1 Details of Training Pipeline and Dataset

In Table 1, we provide details of our training datasets and training strategy. We integrate diverse 2D, 3D, and 4D vision-language datasets for both understanding and generation, which are utilized at different training stages. The entire training pipeline is divided into three stages as follows:

Table 1: Training pipeline and datasets.

| Training Stage | Samples | Data Source |
|---|---|---|
| Stage 1: Fundamental Representation Learning | 1.8 M | ImageNet-1K (Deng et al., 2009), WebVid-10M (Bain et al., 2021), Panda-70M (Chen et al., 2024), InternVid-10M (Wang et al., 2023), Valley (Luo et al., 2023), GranD (Rasheed et al., 2024), ANet-RTL (Huang et al., 2024), MUSE (Ren et al., 2024) |
| Stage 2: Multimodal Spatiotemporal Alignment | 980.6 K | MMScan (Lyu et al., 2024), SCan2Cap (Chen et al., 2021), ScanQA (Azuma et al., 2022), SQA3D (Ma et al., 2022), Multi3dRefer (Zhang et al., 2023), ScanRef (Chen et al., 2020), Chat4d (Zhou & Lee, 2025), CO3D (Reizenstein et al., 2021), Objaverse (Deitke et al., 2023), RealEstate (Zhou et al., 2018), MVImgNet (Yu et al., 2023b) |
| Stage 3: 4D Task Instruction Fine-Tuning | 160.7 K | Chat4D (Zhou & Lee, 2025), 4DNex-10M (Chen et al., 2025), DyCheck (Gao et al., 2022) |

**Stage 1: Fundamental Representation Learning.** This stage is to equip our model with fundamental representation learning capabilities, *i.e.* multi-task visual representation, scene visual representation, and linguistic representation. For the datasets, we integrate typical 2D image/video-text pairs for scene understanding including dense caption and visual QA. The dense caption datasets include ImageNet-1K (Deng et al., 2009), WebVid-10M (Bain et al., 2021), Panda-70M (Chen et al., 2024), InternVid-10M (Wang et al., 2023), and Valley (Luo et al., 2023). The VQA datasets include GranD (Rasheed et al., 2024), ANet-RTL (Huang et al., 2024), and MUSE (Ren et al., 2024). In total, these datasets contain 1.8M samples, where textual captions can also serve as conditional text for scene generation. We use these datasets to preliminarily align the content between visual and linguistic representations for both understanding and generation tasks, which gives our model the ability for fundamental representation learning. Regarding the training strategy, we update the trainable parameters of multiple embeddings, projector, lower layers of the LLM, multi-task heads, while keeping all other modules frozen.

**Stage 2: Multimodal Spatiotemporal Alignment.** This stage aims to further enhance the spatiotemporal awareness of visual and linguistic representations, and transfer the capability of our model to the physical world. For the datasets, we introduce 3D scene understanding datasets, including dense caption, visual QA, and visual grounding. The dense caption datasets consist of MMScan (Lyu et al., 2024) and Scan2Cap (Chen et al., 2021). The visual QA datasets contain ScanQA (Azuma et al., 2022) and SQA3D (Ma et al., 2022). The visual grounding datasets include Multi3DRefer

(Zhang et al., 2023) and ScanRef (Chen et al., 2020). We also introduce a small portion of 4D scene understanding datasets related to dense caption from Chat4D (Zhou & Lee, 2025). Additionally, we integrate 3D generation datasets, such as CO3D (Reizenstein et al., 2021), Objaverse (Deitke et al., 2023), RealEstate10k (Zhou et al., 2018) and MVImgNet (Yu et al., 2023b) to improve the generation quality, where the conditional text can also serve as the caption for scene understanding tasks. In total, these datasets contain 980.6K samples. We employ hybrid datasets to align fine-grained spatiotemporal information between visual and linguistic representations for both understanding and generation tasks. For the training strategy, we update the trainable parameters of spatiotemporal embedding, adaptive cross-attention fusion, higher layers of the LLM, and multi-task heads while keeping the remaining modules frozen.

**Stage 3: 4D Task Instruction Fine-Tuning.** This stage is to improve the generalization of our model for understanding and generation tasks in more complex 4D dynamic scenes. For the datasets, we introduce the typical 4D understanding dataset Chat4D (Zhou & Lee, 2025), and integrate 4D generation datasets including 4DNex-10M (Chen et al., 2025) and DyCheck (Gao et al., 2022). In total, these datasets contain 160.7K samples. We employ these 4D multimodal datasets to perform fine-tuning to adapt our model to the intricacies of complex 4D scenarios. For the training strategy, all trainable parameters are optimized through LoRA adapters. The vision encoder-decoder and geometry encoder remain frozen.

## 2 ABLATION STUDY AND DISCUSSION

### 2.1 APPLICABILITY TO DIFFERENT INPUT SETTINGS

Although we introduce the entire model using multi-view videos as an input example, we also discuss the applicability of our model to other input settings such as single-view video. We use single-view videos collected from the 4D vision datasets of Chat4D (Zhou & Lee, 2025) and the corresponding instruction-following texts as ground truths for evaluation of the understanding task. As shown in Table 2, the performance metrics of our model with single-view videos as input are comparable to our model with multi-view videos as input. This shows that our model is applicable to different input settings and is practical for the real world.

Table 2: Ablation study on the effect of various vision inputs on scene understanding performance.

| Vision input | C↑ | SAcc@0.5↑ | TAcc↑ |
|---|---|---|---|
| w/ Single-view video | 93.6 | 57.9 | 54.5 |
| w/ Multi-view video | **93.8** | **58.2** | **54.6** |

### 2.2 IMPACT OF TASK PROMPT

In Table 3, we analyze the impact of the task prompt on model performance. The results show that introducing the task prompt significantly improves the model performance on both understanding and generation for 4D scenes. The main reason is that the task prompt not only distinguishes the features of understanding and generation within the visual representation, but also guides the subsequent attention mask to dynamically regulate the information flow of different tasks in our model. These mechanisms enhance the capability of our model for multi-task prediction.

Table 3: Ablation study on the effect of task prompt on model performance.

| Task prompt | Chat4D | | | DyCheck | | |
|---|---|---|---|---|---|---|
| | C↑ | SAcc@0.5↑ | TAcc↑ | PSNR↑ | FVD↓ | CLIP-C↑ |
| w/o Prompt | 85.1 | 47.5 | 45.3 | 17.54 | 216.1 | 0.91 |
| w/ Prompt | **93.8** | **58.2** | **54.6** | **21.38** | **152.3** | **0.97** |

## 2.3 Effect of Textual Coordinate Embedding

In Table 4, we study the effect of textual coordinate embedding, *i.e.* pos/time encoding via special token embedding (Li et al., 2025) on understanding performance. First, textual coordinates as instructions improve the fine-grained spatiotemporal understanding of our model. Second, textual coordinate embedding further improves the upper limit of 4D spatiotemporal understanding. This is because textual coordinate embedding helps minimize the risk that large language models misinterpret coordinate values.

Table 4: Discussion on the effect of textual coordinate embedding for scene understanding.

| Text instruction | | C↑ | SAcc@0.5↓ | TAcc↑ |
|---|---|---|---|---|
| w/o Coordinate | | 84.2 | 39.6 | 19.3 |
| w/ Coordinate | w/o Embedding | 90.5 | 55.1 | 52.7 |
| | w/ Embedding | **93.8** | **58.2** | **54.6** |

## 2.4 Limitation and Potential Solution

Despite strong performance in most short-term scene understanding and generation, our Uni4D-LLM struggles with long-term dynamics. Capturing such variations requires memory-based reasoning to model cross-spatiotemporal interactions and causal relations. However, the current framework relies on short-range attention, which primarily captures correlations within limited temporal windows. As a result, the model lacks explicit mechanisms to retain or propagate motion information across extended sequences and thus makes it difficult to reason about long-horizon dependencies. In the future, we plan to integrate a world model (Ha & Schmidhuber, 2018) to enable long-term spatiotemporal reasoning and extend scene understanding and generation to longer temporal horizons. This will improve the practicality of our model.

## 3 Quantitative and Qualitative Results

### 3.1 Quantitative Comparison on 2D Understanding Benchmark

In Table 5, we compare the basic visual understanding capability of our model with other 2D, 3D and 4D VLMs on several typical 2D understanding benchmarks, including VQAv2 (Goyal et al., 2017), MMBench (Liu et al., 2024b), MME (Fu et al., 2024), MM-Vet (Yu et al., 2023a). The results show that our method can maintain the same level of image understanding as other competing VLMs. This also indirectly proves that our model can obtain the initial multimodal understanding capability from the 2D datasets.

Table 5: Quantitative results of VLMs for scene understanding on 2D zero-shot benchmarks.

| | Methods | VQAv2 | MMBench | MME | MM-Vet |
|---|---|---|---|---|---|
| 2D | MobileVLM (Chu et al., 2023) | 47.5 | 59.6 | 1289 | – |
| | Qwen-VL (Bai et al., 2023) | 63.8 | 38.2 | – | – |
| | Qwen-VL-Chat (Bai et al., 2023) | 61.5 | 60.6 | 1488 | – |
| | LLaMA-VID (Li et al., 2024) | – | 65.1 | 1521 | – |
| | LLaVA-1.5 (Liu et al., 2024a) | 58.2 | 65.2 | 1511 | 31.1 |
| 3D | LLaVA-3D (Zhu et al., 2024) | 57.8 | 65.0 | 1502 | 30.9 |
| 4D | Uni4D-LLM (Ours) | 58.8 | 64.5 | 1506 | 30.7 |

### 3.2 Qualitative Comparison on 3D Understanding Benchmark

In Fig. 1, we provide more visual comparisons on 3D scene understanding and generation, where we provide some typical scenes as examples. In 3D understanding, our Uni4D-LLM is more fine-grained than the competing 3D VLMs in representing spatial characteristics of the scene. In 3D generation,

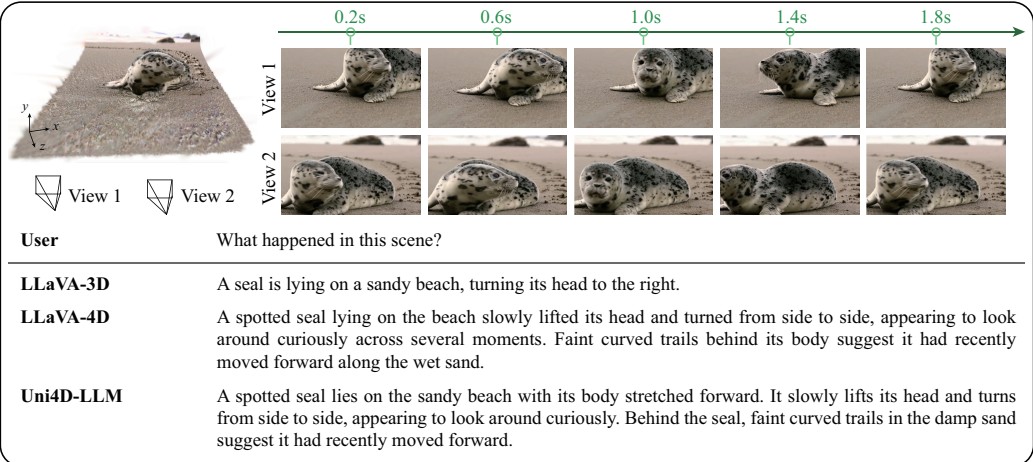

| User | What is the object at [2.16, 1.85, 0.68]? |
|---|---|
| **PQ3D** | A bed located near the wall, horizontally aligned with pillows and a dark blanket on top. |
| **LLaVA-3D** | This is a bed placed at the back of the room, with a dark blanket and pillow, next to the wall and under a window. |
| **Video-3D LLM** | The object is bed covered in dark bedding, placed against the wall and adjacent to a small nightstand and window. |
| **Uni4D-LLM** | The object is a bed covered with dark blue sheets, stationary at the center of the room, aligned with the wall, and surrounded by a nightstand and a desk. |

| User | Please describe this scene. |
|---|---|
| **PQ3D** | The scene contains a bed, sofa, chairs, and a desk with monitors, arranged along the walls of a compact room. |
| **LLaVA-3D** | This indoor scene includes a bed, central desk with monitors, two chairs, and a sofa opposite the bed. |
| **Video-3D LLM** | This is a small bedroom scene with a bed on the right, a central desk with monitors, and a green sofa and chairs on the left. |
| **Uni4D-LLM** | This is a furnished room with a bed beneath the window, a desk with two monitors placed in front, chairs neatly tucked in under the desk, and a green sofa positioned facing the bed across the room. |

(a) Global 3D Scene Understanding    (b) Local 3D Scene Understanding

Figure 1: Visual comparison of VLMs on 3D scene understanding.

| User | What happened in this scene? |
|---|---|
| **LLaVA-3D** | A seal is lying on a sandy beach, turning its head to the right. |
| **LLaVA-4D** | A spotted seal lying on the beach slowly lifted its head and turned from side to side, appearing to look around curiously across several moments. Faint curved trails behind its body suggest it had recently moved forward along the wet sand. |
| **Uni4D-LLM** | A spotted seal lies on the sandy beach with its body stretched forward. It slowly lifts its head and turns from side to side, appearing to look around curiously. Behind the seal, faint curved trails in the damp sand suggest it had recently moved forward. |

Figure 2: Visual comparison of VLMs on 4D scene understanding.

the results generated by our model are on par with those of 3D generation models. These results demonstrate the effectiveness of our model on both 3D scene understanding and generation.

### 3.3 QUALITATIVE COMPARISON ON 4D UNDERSTANDING AND GENERATION BENCHMARK

In Fig. 2 and 3, we also provide more visual comparisons on 4D scene understanding and generation. In 4D understanding, 3D VLMs struggle to capture temporal dynamics, while our model demonstrates strong spatiotemporal reasoning on par with recent 4D VLMs. In 4D generation, our Uni4D-LLM produces sharp and coherent results that rival those of advanced 4D diffusion models. These results demonstrate the superiority of our Uni4D-LLM in 4D understanding and generation, underscoring its potential as a unified multi-task framework for the physical world.

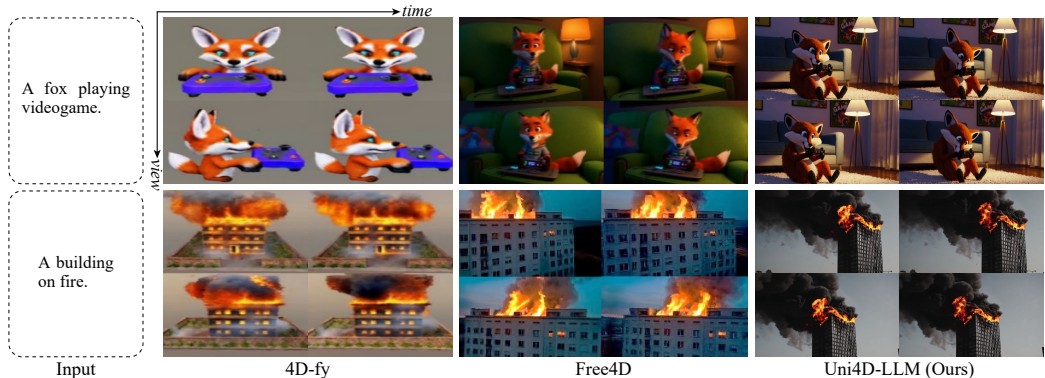

Figure 3: Visual comparison on 4D scene generation.