# OpenReview forum: "Uni4D-LLM: A Unified SpatioTemporal-Aware VLM for 4D Understanding and Generation"
_ICLR.cc/2026/Conference — ICLR 2026 Conference Withdrawn Submission_

### Official Review · Reviewer_PYh6 · 2025-10-27

**Soundness:** 3
**Presentation:** 3
**Contribution:** 2
**Rating:** 6
**Confidence:** 3

**Summary:**

This paper proposes Uni4D-LLM, a unified VLM framework which is capable of performing both 4D understanding and generation. Uni4D-LLM achieves this by utilizing a shared spatiotemporal-aware representation that incorporate semantic, appearance, and geometric features. Also, a shared Transformer-based architecture enables both understanding and diffusion-based generation tasks to be possible under a single framework. Experimental results demonstrate that the proposed Uni4D-LLM achieves comparable or superior results on scene understanding and generation compared to the existing models.

**Strengths:**

- This paper proposes the first (as far as I know) unification of a 4D foundation model on both 4D scene understanding and generation, which could serve as the pioneering exploration of this direction.

- I think the idea of injecting 4D geometric features using MonST3R is kind of interesting to me. I think most 3D/4D-LLMs are focusing on how to inject semantic features into LLMs, but have largely overlooked the geometric features.

- The ablation study in this paper is comprehensive and informative, which can provide valuable guidance on how to train and use the unified 4D-LLM model.

- The model can be trained with only 8 RTX 4090 GPUs, which seems to be computationally efficient, especially comparing with other 3D/4D-LLMs that require massive training resources. (Additional questions about this point in the "Questions" part)

**Weaknesses:**

- The core idea of this paper is to propose a unified 4D foundation model that can perform both understanding and generation. However, from the experimental results, it seems that the performance gets a bit dropped when optimizing the two types of tasks together. It indicates that there is not so much mutual benefit when unifying understanding and generation in a single model. In this case, the most notable benefit of the unification of understanding and generation would be parameter efficiency, as shown in Figure 7, with a little compromise in performance. Therefore, I find it not very justified to have foundational benefit from the task optimization perspective.

- The settings of the ablations on spatiotemporal embedding are not clear. From Figure 2, the geometry encoder (MonST3R in the paper) is the one that produces spatiotemporal embedding. MonST3R is an encoder that encodes dynamic videos to poses and positions, then what does it mean by "w/ Spatial"? Are we replacing MonST3R with some degraded version like DUSt3R? Or MonST3R encodes a static stack of only the first frame of the video? The settings of this ablation study needs to be clarified.

**Questions:**

- As mentioned in "Weaknesses", there is no strong justification for designing a unification of understanding and generation, as there is no obvious sign of mutual benefit between these two streams of tasks. One potential application that needs this kind of unification would be 4D scene editing, or 4D scene generation with multi-round user interaction. Is this model capable of doing this task? If so, this could be some justification that supports the need of this kind of unified model for 4D understanding and generation.

- The whole model is trained on only 8 RTX 4090 GPUs that seems to be not very demanding for a 4D foundation model, which is great. However, I cannot find in the paper how long / how many iterations does it take to train the model. Therefore, it becomes unclear whether the training process of this model is indeed efficient or not.

---

### Official Review · Reviewer_FEpW · 2025-10-31

**Soundness:** 3
**Presentation:** 2
**Contribution:** 3
**Rating:** 6
**Confidence:** 5

**Summary:**

The paper proposes Uni4D-LLM, a single transformer that unifies 4D scene understanding (autoregressive text prediction) and generation (4D diffusion). The paper achieves this by (1) building a shared spatiotemporal visual representation that includes high-level semantic features, noise-injected appearance features, and 4D geometric features , and (2) using both autoregressive and diffusion heads on a shared transformer backbone with task-specific attention masks. The model is trained in three stages with instruction tuning. The model is competitive with state-of-the-art VLMs for understanding and approaches the best diffusion models for generation, especially when paired with Gaussian Splatting (GS).

**Strengths:**

1.	It is the first to unify 4D scene understanding (autoregressive) and generation (diffusion) within a single, principled framework. This is achieved through a shared spatiotemporal representation and a single Transformer using task-specific attention masks, moving beyond prior fragmented approaches.

2.	The method is well-designed and rigorously validated with thorough ablation studies that confirm the value of each component (e.g., spatiotemporal embedding, attention masks). The paper uses clear figures and descriptions to make the complex architecture easy to understand.

3.	The model achieves highly competitive, state-of-the-art results on both understanding and generation benchmarks.

**Weaknesses:**

1.	While the paper presents the first unified model for 4D understanding and generation, the technical approach could be perceived as a straightforward combination of existing unification architectures (from 2D/3D) and established 4D representation methods.

2.	The core motivation for unifying 4D understanding and generation is not fully convincing. The experiments do not clearly demonstrate a synergistic relationship where the two tasks mutually benefit each other within the unified model. Since the model's performance is slightly below the state-of-the-art for specialized understanding or generation models, the paper should more explicitly justify the benefits of unification.

3.	It’s better to include more comparisons with more recent state-of-the-art methods and newer benchmarks (e.g., VSI-Bench).

4.	Several key details necessary for reproducibility and a full assessment of the work are missing. The paper would be significantly improved by including more information on the training process, such as the total training time, the specific format of the training datasets, and the procedure used to construct or preprocess the data.

5.	Typo: Line 369: RealEstate10 -> RealEstate10k

**Questions:**

1.	During training, are the semantic encoder, the main model weights, and the VAE weights from Wan2.1 loaded into VRAM simultaneously? Could you provide an estimate of the total VRAM consumption required for the training setup described?

2.	How does the shared Transformer backbone differentiate between an understanding task and a generation task at inference? Furthermore, since the input representations are task-specific (e.g., semantic features for understanding vs. noisy appearance features for generation), what would be the hypothetical output if the tokens from an understanding task were processed by the Diffusion Head?

3.	Benchmarks like Scan2Cap, Multi3DRefer, and ScanRef require grounding objects in an absolute global coordinate system. However, the model's geometry encoder, MonST3R, predicts relative camera poses and scene positions. How does the model align its internally predicted relative coordinates with the absolute coordinates of specific objects during evaluation on these grounding tasks?

4.	For NVS datasets such as RealEstate10K and DyCheck, generation is conditioned on specific camera parameters. How does the model incorporate this camera conditioning to generate consistent novel views corresponding to given camera poses?

5.	The results show that adding GS as a post-processing step improves generation quality. Could you provide a formal, step-by-step description of how the model's output is integrated with GS to achieve this enhancement?

6.	What is the number of diffusion steps T used during inference at test time? Additionally, what is the end-to-end latency for generation tasks (e.g., text-to-4D and image-to-4D) at typical resolutions and view/time counts?

---

### Official Review · Reviewer_ym1y · 2025-11-02

**Soundness:** 2
**Presentation:** 2
**Contribution:** 1
**Rating:** 2
**Confidence:** 4

**Summary:**

The paper proposes Uni4D-LLM, a unified vision-language model for 4D (spatiotemporal) scene understanding and generation. It aims to bridge autoregressive (AR) and diffusion paradigms within a single Transformer architecture. The approach integrates: Spatiotemporal-aware visual representation, combining semantic, appearance/noise, and 4D geometric features via adaptive cross-attention; Unified hybrid LLM architecture, using shared Transformer weights with task-specific heads and attention masks for AR and diffusion tasks; Multi-task optimization, jointly aligning visual and linguistic tokens with instruction-tuned training across 2D, 3D, and 4D datasets. The paper claims state-of-the-art or competitive performance on 3D/4D benchmarks for both understanding and generation

**Strengths:**

1, Unifying 4D scene understanding and generation is an important step beyond fragmented 3D/4D pipelines.

2, The idea of using attention masks within a single Transformer to emulate both AR and diffusion paradigms is elegant and theoretically consistent.

3,Evaluation spans multiple 3D and 4D benchmarks, including ablations on fusion strategies, attention masks, and spatiotemporal embeddings, showing incremental improvements.

4,The paper is well structured, and the visual diagrams effectively convey the proposed unification.

**Weaknesses:**

1, The claimed “first unified 4D VLM” contribution is largely a system-level integration of existing ideas (e.g., autoregressive-diffusion unification from MetaQueries, BLIP3O, Show-O or VILA-U; spatiotemporal embedding from MonST3R; cross-attention fusion from standard multimodal LLMs). The novelty lies more in the engineering combination than in new algorithmic insight.

2, In Stage 1, the paper claims to use multiple large-scale 2D image and video-text datasets such as ImageNet-1K, WebVid-10M, Panda-70M, InternVid-10M, and Valley, but reports that only 1.8M samples are actually used for training. It remains unclear how these “good pairs” are selected from the much larger datasets—whether any filtering, quality assessment, or weighting strategy is applied to ensure balanced coverage across sources. Moreover, the paper does not explain how images and videos are jointly used during training.

3, Some training details are not provided: e.g., the training time used for each stage.

**Major Concern:** Implausible 4D Generation and Unsubstantiated Experimental Claims

The claimed 4D generation capability of Uni4D-LLM is highly questionable given the reported computational setup and vague experimental reporting. According to the paper, all experiments were conducted on only 8× RTX 3090 GPUs, which is severely insufficient for such a resource-intensive task. The model performs video-to-video (4D) training, where both input and output are temporal sequences—requiring large-scale visual encoding, spatiotemporal reasoning, and diffusion-based generation within a unified architecture.

From my experience, even standard unified video generation and understanding models typically require at least 32× A100 GPUs to achieve stable convergence and acceptable quality. Extending this to 4D generation, which further incorporates spatial-temporal geometry and multi-view consistency, dramatically increases computational demands. Achieving such performance with only 8×3090 GPUs appears highly implausible.

Moreover, the proposed two-stage training strategy—representation alignment followed by LoRA fine-tuning of the LLM—is far too lightweight to support a full-scale diffusion model. LoRA adaptation on a frozen LLM backbone cannot plausibly generate high-quality latent features to drive a heavy decoder such as Wan’s VAE. In effect, the paper attempts to fit a massive 4D generative system with an extremely small number of trainable parameters, which is almost infeasible in practice.

In addition, the generation stage lacks explicit view or spatial conditioning in the prompt. Without multi-view or geometric cues, it is theoretically unclear how the model could produce spatially and temporally coherent multi-view outputs. From practical experience in generative modeling, such setups (where training data contain view-varying videos without explicit conditioning) are notoriously unstable and rarely converge.

The 4D video samples shown in Figure 2 appear almost impossible to obtain under the described configuration. Achieving consistent and realistic 4D video synthesis on such limited compute using only LoRA-based fine-tuning is not credible without massive undisclosed pretraining or external priors—neither of which are clarified in the paper.

Beyond computational issues, the paper lacks quantitative evaluation (e.g., FVD, FID, CLIP-Score) to objectively validate video generation quality. Only few qualitative visuals are shown in supplementary (even not looks good). The authors also provide no details on dataset size, sequence length, or resolution, making reproducibility difficult. Finally, temporal and geometric consistency metrics—which are essential for any 4D model—are entirely missing.

Taken together, the combination of insufficient compute, vague training design, missing quantitative evaluation, and lack of reproducibility details makes the claimed 4D generation results—especially those in Figure 2—highly implausible and scientifically unsubstantiated.

**Questions:**

See weakness

---

### Note · Authors · 2025-11-14

I have read and agree with the venue's withdrawal policy on behalf of myself and my co-authors.